Accepted at the ICLR 2024 Workshop on AI4Differential Equations In Science

# MECHANISTIC NEURAL NETWORKS FOR SCIENTIFIC MACHINE LEARNING

**Adeel Pervez**
Informatics Institute
University of Amsterdam
Amsterdam, The Netherlands
a.a.pervez@uva.nl

**Francesco Locatello**
Institute of Science and Technology
Klosterneuburg, Austria
Francesco.Locatello@ist.ac.at

**Efstratios Gavves**
Informatics Institute
University of Amsterdam
Amsterdam, The Netherlands
e.gavves@uva.nl

## ABSTRACT

This paper presents *Mechanistic Neural Networks*, a neural network design for machine learning applications in the sciences. It incorporates a new *Mechanistic Block* in standard architectures to explicitly learn governing differential equations as representations, revealing the underlying dynamics of data and enhancing interpretability and efficiency in data modeling. Central to our approach is a novel fast, parallel and scalable *Relaxed Linear Programming Solver* (NeuRLP) using a differentiable optimization approach for ODE learning and solving. Mechanistic Neural Networks demonstrate their versatility for scientific machine learning applications on tasks from equation discovery to dynamic systems modeling. [1]

## 1 INTRODUCTION

In this paper, we introduce *Mechanistic Neural Networks*, a new neural network design that contains one or more *Mechanistic Block* that explicitly integrate governing equations as symbolic elements in the form of ODE representations. To efficiently train them, we revisit classical results on linear programs (Young, 1961; Rabinowitz, 1968) and develop a GPU-friendly solver. Together, they enable automating modeling and discovery of best-fitting mechanisms from data in an efficient, scalable, and interpretable way.

Mechanistic networks are composed of two parts: a mechanistic encoder and a solver. The output of the mechanistic encoder is an explicit symbolic "*ODE representation*" $\mathcal{U}_x$ of the general form

$$\mathcal{U}_x : F(\alpha, x) = 0, \qquad (1) \qquad \text{where } \alpha = f_\theta(x) \qquad (2)$$

$\mathcal{U}_x$ is a family of ordinary differential equations $F(\alpha, x) = 0$, governed by learnable coefficients $\alpha$ that may be time-dependent. Coefficients $\alpha$ are obtained from the mechanistic encoder $f_\theta$, and parameters $\theta$ are trained to optimally model the evolution of data $x = [x_1, ..., x_t]$ over time.

When training ODE representations we must simultaneously *learn* the precise form of multiple independent ODEs (or independent systems of ODEs) and *solve* them over several time steps. Sequential numerical solvers such as Runge-Kutta used in Neural ODEs (Chen et al., 2018) are simply too inefficient for solving large batches of independent ODEs as required for Mechanistic Neural Networks.

With Mechanistic Neural Networks, we address both challenges directly in a native neural network context, as shown in Figure 1. We propose a novel neural *Relaxed Linear Programming Solver* (NeuRLP) for ODEs based on differentiable optimization methods (Amos and Kolter, 2017; Young,

---

[1]Source code is available at https://github.com/alpz/mech-nn

| | Neural ODE,UDE Chen et al. (2018) Rackauckas et al. (2020) | SINDy Brunton et al. (2016) | Neural Operators Li et al. (2020c) | Mech. NN |
|---|---|---|---|---|
| Linear discovery | – | ✓ | – | ✓ |
| Nonlinear discovery | – | – | – | ✓ |
| Physical parameters | ✓ | ✓ | – | ✓ |
| Forecasting | ✓ | – | ✓ | ✓ |
| Interpretability | – | ✓ | – | ✓ |

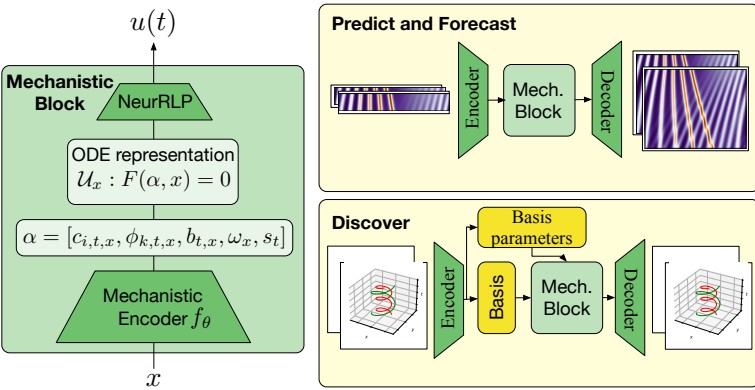

Figure 1: Mechanistic Neural Networks are a new neural network design that learn explicit ODE representations for prediction and discovery.

1962). NeuRLP has three critical advantages over traditional solvers, leading to more efficient learning over longer sequences than traditional sequential solvers. These are: (i) *step parallelism*, i.e., being able to solve for hundreds of ODE time steps in parallel, allowing for faster solving and efficient gradient flow; (ii) *batch parallelism*, where we can solve in parallel on GPU batches of independent systems of ODEs in a single forward pass; (iii) *learned step sizes*, where the step sizes are differentiable and learnable by a neural network. This makes the NeurLP solver ideal for training efficiently neural networks that model complex dynamic systems.

**Relevance for scientific applications.** Machine Learning for dynamical systems has adopted specialized methodologies such Physics-Informed Neural Networks (Raissi et al., 2019) for solving PDEs, neural operators (Li et al., 2020c) for prediction, linear regression with basis functions for discovery (Brunton et al., 2016), Neural ODEs (Chen et al., 2018) for dynamical systems. Being able to weave in governing equations in neural representations and solve them efficiently, Mechanistic Neural Networks potentially offer a stepping stone for broad scientific application of machine learning (Figure 1). We demonstrate this with experiments on tasks from each of above settings.

## 2 MECHANISTIC NEURAL NETWORKS

Formally, a Mechanistic Network contains a mechanistic encoder in a mechanistic block that takes an input $x$ and generates a differential equation $\mathcal{U}_x$ as representation according to equations 2 and 1. The family of ordinary differential equations $\mathcal{U}_x : F(\alpha, x) = 0$

$$\overbrace{\sum_{i=0}^{d} c_i(t; x)u^{(i)}}^{\text{Linear Terms}} + \overbrace{\sum_{k=0}^{r} \phi_k(t; x)g_k(t, u, u', \ldots)}^{\text{Non-Linear Terms}} = b(t; x), \tag{3}$$

represents a broad parameterization for the ODE representation, with an arbitrary number $d$ of linear terms with derivatives $u^{(i)}$ and an arbitrary number $r$ of nonlinear terms $g_k$ including derivatives $u^{(k)}, k = 1, ..., d$. The coefficients $c_i(t; x), \phi_k(t; x)$ for the linear and nonlinear terms are functions that possibly depend on the time variable $t$ (thus non-autonomous ODEs), and on input $x$.

After computing the ODE representation $\mathcal{U}_x$, we solve it with our specially designed parallel solver *NeuRLP* for $n$ time steps and get a numerical solution as output of the mechanistic block: $z = \texttt{solve\_ode}(\mathcal{U}_x, \omega_x, n)$. $\omega_x$ includes initial or boundary conditions and variables controlling step sizes that may be specific to the input $x$. $\omega_x$ can be learned by NeurLP, unlike traditional solvers.

Equations 2– 3 provide the mathematical description of ODE representation in mechanistic blocks in continuous time. To implement them in a neural network we discretize the continuous coefficients, parameters, function values, and derivatives in the ODE 3,

$$\sum_{i=0}^{d} c_{i,t,x} u_t^{(i)} + \sum_{k=0}^{r} \phi_{k,t,x} g_k(t, u_t, u_t', \ldots)) = b_{t,x}, \text{ s.t. } [u_{t=1}, u_{t=1}', \ldots] = \omega_x \tag{4}$$

at discrete times $t = 1, \ldots, n$ and with $n - 1$ time steps $s_t$. Steps $s_t$ do not have to be uniformly equal, and can either be a hyperparameter or learned. Similarly, other conditions in $\omega_x$ can be a hyperparameter or learned to best explain the data evolution. In the general case, we learn $s_t$ and $\omega_x$ and parameterize all coefficients of ODE representation $\alpha = [c_{i,t,x}, \phi_{k,t,x}, b_{t,x}, s_t, \omega_x]$ with a standard network $f_\theta(x)$ (2). Coefficients $\alpha$ are obtained with a single forward pass for times $t = 1, \ldots, n$. During training, we need to compute gradients $\frac{\partial f}{d\alpha}, \frac{\partial f}{d\theta}$ through the ODE solver. This is an expensive operation for large systems which calls for an efficient, neural-friendly ODE solver.

## 3 NEURAL RELAXED LINEAR PROGRAMMING ODE SOLVER

We present the *Neural Relaxed Linear Programming (NeuRLP) solver*, a novel and efficient algorithm for solving batches of independent ODEs. We show how to solve linear ODEs with differentiable quadratic programming with equality constraints motivated from a linear programming method (Young, 1961) for linear ODEs.

We start with discretized linear ODEs ignoring the nonlinear terms $g_k$ in equation 4, that is $\sum_{i=0}^{d} c_{i,t,x} u_t^{(i)} = b_{t,x}$, s.t. $[u_{t=1}, u_{t=1}', \ldots] = \omega_x$. One can solve linear ODEs by solving corresponding linear programs of the form

$$\begin{aligned} \text{minimize} \quad & \delta^\top z \\ \text{subject to} \quad & Az \geq \beta, \end{aligned} \tag{5}$$

where $z$ is the variable that we optimize for and $A \in \mathbb{R}^{m \times n_v}$ and $\beta \in \mathbb{R}^m$ represent the (inequality or equality) constraints and $\delta \in \mathbb{R}^{n_v}$ represents the cost of each variable. In the following we describe the constraints, variables and the optimization objective.

We have three types of constraints: the equality constraints that define the ODE itself, initial value constraints, and smoothness constraints for the solution of the linear program.

**ODE equation constraints** specify that at each time step $t$ the left-hand side of the discretized ODE is equal to the right-hand side, *e.g.*, for a second-order ODE,

$$c_{2,t} u_t'' + c_{1,t} u_t' + c_{0,t} u_t = b_t, \forall t \in \{1, \ldots, n\}. \tag{6}$$

**Initial-value constraints** specify constraints on the function or its derivatives for the initial conditions at $t = 1$, *e.g.*, that they have to be equal to 0, as $u_1 = 0, \quad u_1' = 0$.

**Smoothness constraints** ensure the solutions of the linear program to the function and derivative values at each time step are $\epsilon$-close in neighboring locations. We determine the values in neighboring locations by Taylor approximations up to error $\epsilon$. We define one Taylor approximation for the forward-time evolution of the ODE, $t : 1 \to n$, and one for the backward-time, $t : n \to 1$. If we are interested in a second-order ODE for instance, we have as Taylor expansions, with $\epsilon \geq 0$:

$$\left. \begin{aligned} |u_t + s_t u_t' + \tfrac{1}{2} s_t^2 u_t'' - u_{t+1}| &\leq \epsilon \\ |s_t u_t' + \tfrac{1}{2} s_t^2 u_t'' - s_t u_{t+1}'| &\leq \epsilon \end{aligned} \right\} \text{Forward} \qquad \left. \begin{aligned} |u_t - s_t u_t' + \tfrac{1}{2} s_t^2 u_t'' - u_{t-1}| &\leq \epsilon \\ |-s_t u_t' + \tfrac{1}{2} s_t^2 u_t'' + s_t u_{t-1}'| &\leq \epsilon \end{aligned} \right\} \text{Backward}$$

$$\tag{7} \qquad\qquad\qquad\qquad\qquad\qquad \tag{8}$$

**Variables.** In $z$ we introduce three types of variables. First, we introduce per time step $t \in \{1, \ldots, n\}$ one variable $z_t^0$ that corresponds to the value of the function at time $t$, that is $u_t$. Second, we introduce per time step $t \in \{1, \ldots, n\}$ one variable $z_t^i$ that corresponds to the value of the $i$-th function derivative at time $t$ for all derivative orders, that is $u_t^{(i)}, i = 1, \ldots, d$. Third, we introduce a single scalar variable $\epsilon$ shared for all time steps that corresponds to the error of the Taylor approximation for all function values and derivatives.

**Optimization objective** The objective of the linear program is to minimize the smoothness error $\epsilon$. Solving the linear program, we obtain in $z$ the function values and derivatives that satisfy all the ODE equality and inequality constraints, including $\epsilon$-smoothness.

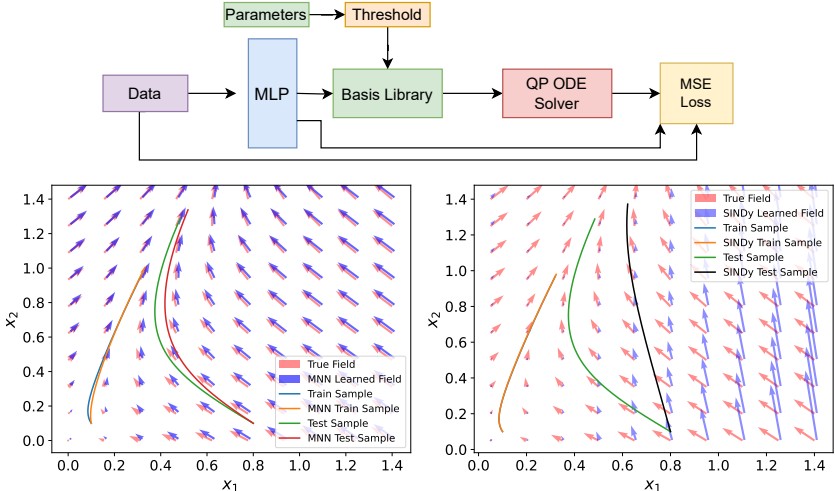

Figure 2: ODE discovery architecture (top) and learned ODE vector fields for MNN (bottom left) and SINDy (bottom right) with non-linear tanh.

**Efficient Quadratic relaxation and extension.** Solving ODEs using the LP method inside neural networks has three main obstacles: 1) the solutions to the LP are not continuously differentiable (Wilder et al., 2019), 2) solving linear programs is generally done using specialized solvers that do not take advantage of GPU parallelization and are too inefficient for neural networks applications, and 3) The matrices $A$ are highly sparse where dense methods (such as from Amos and Kolter (2017)) are infeasible for large problems. We solve these problems by reframing the problem as an equality constraint-only quadratic program with regularization to ensure boundedness and the use of sparse methods for large problems. We cover further details including error analysis, complexity and numerical validation in A.4, A.2, A.3. We also extend the LP method to nonlinear ODEs by combining learning with solving in A.6. The presentation above applies the quadratic approximation to the linear program 5. The same approximation can also be applied to the dual of 5 and in practice can lead to a more efficient implementation.

## 4 EXAMPLE APPLICATIONS IN SCIENTIFIC ML

We benchmark Mechanistic Networks in *five different settings* including discovery of equations, PDE solving, , PDE solving, $n$-body prediction, physical parameters discovery and time series forecasting (with the last two in the appendix) from scientific ML applications.

**ODE Discovery.** Gold standard in discovering governing equations is SINDy (Brunton et al., 2016; Rudy et al., 2017) which models the problem as linear regression on a library of candidate nonlinear basis functions $\Theta(x)$. SINDy is constrained to problems where the governing equations are linear combinations of (nonlinear) basis functions. Similar to Brunton et al. (2016), we model nonlinear ODEs $\frac{d}{dt}\mathbf{x}(t) = \mathbf{F}(\mathbf{x}(t))$, from equation 1 with polynomial basis but followed by a further nonlinear transformation depending on the problem (see B.3,C.1 for details). We experiment with the following ODE systems: (1) the Lorenz system, and (2) ODEs with complex nonlinear function of the form $\frac{d}{dt}\mathbf{x}(t) = F(A\mathbf{x}(t))$, where $A$ is a linear transformation and $F$ is a nonlinear function such as tanh, and (3) ODEs with rational function derivatives, $\frac{d}{dt}\mathbf{x}(t) = \frac{p(x)}{q(x)}$, where $p$ and $q$ are

Table 1: Solving 1d KdV(Brandstetter et al., 2022) with N train samples.

| Method | RMSE | |
|---|---|---|
| | N=512 | N=256 |
| ResNet | 0.0223 | 0.0392 |
| ResNet-LPSDA-1 | 0.0200 | 0.0284 |
| ResNet-LPSDA-2 | 0.0111 | 0.0185 |
| ResNet-LPSDA-3 | 0.0155 | 0.0269 |
| ResNet-LPSDA-4 | 0.0113 | 0.0184 |
| FNO | 0.0276 | 0.0407 |
| FNO-LPSDA | 0.0055 | 0.0132 |
| FNO-AR | 0.0030 | 0.0058 |
| FNO-AR-LPSDA | 0.0010 | 0.0037 |
| Mechanistic NN (50 sec) | 0.0039 | 0.0086 |

polynomials. ODEs with non-linear functions cannot be modeled by the approach employed by SINDy. Results are shown in Figure 2 and Figures 6 and 7 in the appendix. For the Lorenz system which can be described as linear basis combinations, both SINDy and variants, as well as Mechanistic NNs recover the exact equations. For complex nonlinear and rational function ODEs which requires

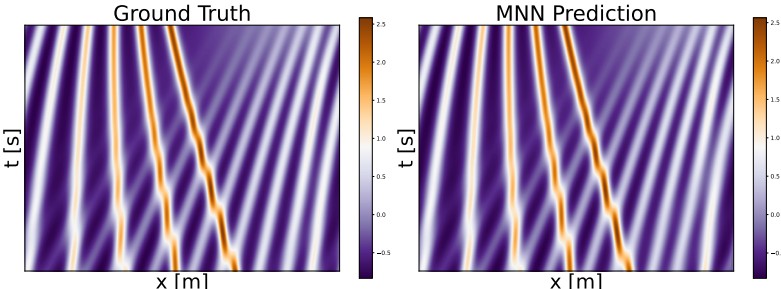

Figure 3: Solving 1d KdV: comparison of ground truth and MNN prediction for 100 seconds.

nonlinear functions of basis combinations, SINDy exhibits poor generalization and overfits to the training domain. See appendices B.3, C.1 for more details and discovered equations.

**PDE Solving.** FNO (Li et al., 2020c) and Lie-group augmented models (Brandstetter et al., 2022) are strong state-of-the-art baselines for PDE solvers. FNO models are deep operator architectures whose intermediate layers perform spectral operations on the input. Lie-group augmentations for PDEs exploit that PDEs conform by definition to certain Lie symmetries to generate new training data.

We adapt Mechanistic NNs from ODEs to PDEs. For 1-d PDEs, we simply model spatial dimensions with *independent* ODEs. With a spatial dimension of 256 and prediction over 10 time steps, we learn 256 ODEs for 10 time steps each. For 2-d PDEs we use a neural operator model with stacked MNN layers. We provide further details of the model and training and visualizations in appendix C.5. Following Brandstetter et al. (2022), we compare relative MSE loss using Lie-symmetry augmented ResNet, FNO and autoregressive FNO on the 1-d KdV equation (Figure 3) and with FNO on 2-d Darcy Flow (Li et al., 2020c) (Table 2 in the appendix).

We use 50 second 1-d KdV equation data and predict for 100 steps. We use 10 time steps of history as opposed to the baselines which use 20 time steps. We also show visualizations for KdV prediction in Figure 3 on a 100 sec dataset. Mechanistic NNs are competitive with FNO and augmented models without using any specialized adaptions for stable rollout (Brandstetter et al., 2022).

**N-body prediction.** The task is to predict future locations and velocities of all bodies in a system given past observations of locations and velocities. We use Neural ODEs (Chen et al., 2018; Norcliffe et al., 2020) as a gold standard baseline and an MNN with second-order ODEs for our model. We use planetary ephemerides data from the JPL Horizons database for solar system dynamics (Giorgini, 2015). The data is positions and velocities for the 25 largest bodies in the solar system from 1980 to 2015 with a step size of 12 hours. We use the first 70% of the data for training and the rest for evaluation. At training the MNN model predicts the next 50 steps given 50 input steps. At testing we rollout predictions for 2000 steps given the starting 50 steps. See prediction rollouts for Earth and Mars

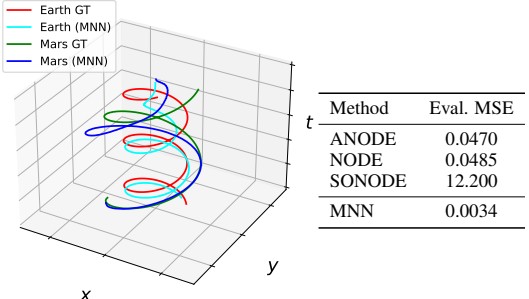

| Method | Eval. MSE |
|--------|-----------|
| ANODE  | 0.0470    |
| NODE   | 0.0485    |
| SONODE | 12.200    |
| MNN    | 0.0034    |

Figure 4: Ephemerides experiment predictions for orbits of Earth, Mars (left) for 1000 days (2000 steps) and eval loss (right). Showing x,y coordinates with time for visualization.

in and evaluation losses in Figure 4. Mechanistic NNs improve Neural ODEs significantly by at least a factor of 10.

## 5  CONCLUSION

Mechanistic Neural Networks (MNNs) are an approach for modeling complex dynamical and physical systems in terms of explicit governing mechanism. MNNs represent the evolution of complex dynamics in terms of families of differential equations making them flexible and able to model the dynamics of complex systems combined with a specialized solver. We demonstrate the effectiveness of the method with experiments in diverse settings.

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

## A    FURTHER DETAILS FOR SECTION 3

### A.1    LINEAR PROGRAMS

A linear program in the primal form is specified by a linear objective and a set of linear constraints.

$$\begin{aligned} \text{minimize} \quad & c^t x \\ \text{subject to} \quad & Ax = b, \\ & x \geq 0 \end{aligned} \tag{9}$$

where $A \in \mathbb{R}^{m \times n}$, $c \in \mathbb{R}^n$, $b \in \mathbb{R}^m$ the following specifies a linear program. Matrix $A$ and vector $b$ define the equality constraints that the solution for $x$ must comply with. $c^t x$ is a cost that the solution $x$ must minimize. The linear program can also be written in dual form,

$$\begin{aligned} \text{minimize} \quad & b^t \lambda \\ \text{subject to} \quad & A^t \lambda = c. \end{aligned} \tag{10}$$

### A.2    ERROR ANALYSIS

We consider the case of a second order linear ODE with an $N$-step grid. For simplicity we consider a fixed step size $h$, i.e., $s_t = h$.

$$c_2 u'' + c_1 u' + c_0 u = b, \tag{11}$$

Let $u(t)$ denote the true solution with initial conditions $u(0) = r$, $u'(0) = s$.

Define

$$\tilde{u}_{t+1} = u_t + hu'_t + \frac{1}{2} h^2 u''_t, \tag{12}$$

$$\tilde{u}'_{t+1} = hu'_t + \frac{1}{2} h^2 u''_t, \tag{13}$$

as Taylor approximations and $\tilde{u}''_t$ is obtained by plugging the approximate values in the ODE 11.

We consider the following Taylor constraints (expressions 7 8) for the function and its first derivative. We use the absolute-value error inequalities for conciseness, the case for equalities is similar.

$$|\tilde{u}_{t+1} - u_{t+1}| \leq \epsilon \tag{14}$$

$$|h\tilde{u}'_{t+1} - hu'_{t+1}| \leq \epsilon \tag{15}$$

**Step $t = 1$.**    From Taylor's theorem we have that for the first step, $t = 1$,

$$u(h) = \tilde{u}_1 + O(h^3) \tag{16}$$

$$u'(h) = \tilde{u}'_1 + O(h^2) \tag{17}$$

From 14, 15

$$u_1 = \tilde{u}_1 + O(\epsilon + h^3) \tag{18}$$

$$u'_1 = \tilde{u}'_1 + O(\epsilon/h + h^2) \tag{19}$$

This implies a local error at each step of $O(\epsilon + h^3)$ in $u_t$.

**Step $t = 2$.**    To estimate the error at step 2 we need to estimate the error in $u''_1$ at step 1.

For $u''_1$ we get the error by multiplying the error in $u_1$ by $\frac{c_0}{c_2}$ and that of $u'_1$ by $\frac{c_1}{c_2}$ and adding.

$$u''_1 = \tilde{u}''_1 + O(\frac{c_1}{c_2}(\frac{\epsilon}{h} + h^2)) + O(\frac{c_0}{c_2}(\epsilon + h^3)) \tag{20}$$

Notice that $u''_1$ always appears with a coefficient of $h^2$. Assuming $\frac{c_0}{c_2}$ is $O(\frac{1}{h^2})$ and $\frac{c_1}{c_2}$ is $O(\frac{1}{h})$ we have

$$h^2 u''_1 = h^2 \tilde{u}''_1 + O(\epsilon + h^3) + O(\epsilon + h^3). \tag{21}$$

Each of the terms $u_1, hu'_1, h^2 u''_1$ contribute an error of $O(\epsilon + h^3)$ to $u_2$ plus an additional error of $O(h^3)$ arising from the Taylor approximation and an error of $\epsilon$ arising from the inequalities 14, 15.

$N$ **Steps.**  Proceeding similarly, after $N$ steps we get a cumulative error of $O(N(\epsilon + h^3))$.

For $\epsilon \approx h^3$ and $N \approx 1/h$, we get an error of $O(h^2)$ for $N$-steps under the assumption that $\frac{c0}{c2}$ is $O(\frac{1}{h^2})$ and $\frac{c1}{c2}$ is $O(\frac{1}{h})$.

The analysis implies that the cumulative error can become large for equations where $\frac{c0}{c2}$, $\frac{c1}{c2}$ are large. Or, in little omega notation, $\frac{c0}{c2}$ is $\omega(\frac{1}{h^2})$ and $\frac{c1}{c2}$ is $\omega(\frac{1}{h})$ .

### A.3  EFFICIENT QUADRATIC RELAXATION

Solving ODEs using the LP method inside neural networks has three main obstacles: 1) the solutions to the LP are not continuously differentiable (Wilder et al., 2019) with respect to the variables $A, b, c$ that interest us and 2) solving linear programs is generally done using specialized solvers that do not take advantage of GPU parallelization and are too inefficient for neural networks applications, and 3) The matrices $A$ are highly sparse where dense methods for solving and computing gradients (such as from Amos and Kolter (2017)) are infeasible for large problems.

We can avoid the non-differentiability of linear programs by including a diagonal convex quadratic term (Wilder et al., 2019) as a regularization term, converting inequalities into equalities by slack variables and removing non-negativity constraints (Pervez et al., 2023) to obtain an equality-constrained quadratic program,

$$\begin{array}{ll} \text{minimize} & \frac{1}{2}z^\top G z + \delta^\top z + \xi^\top \xi \\ \text{subject to} & Az = \beta + \xi, \end{array} \tag{22}$$

where $G = \gamma I, \gamma \in \mathbb{R}$ is a multiple of the identity for a relaxation parameter $\gamma$ and $\xi$ are slack variables. Importantly, equality-constrained quadratic programs can be directly and very efficiently solved in parallel on GPU (Pervez et al., 2023). This is why we rewrite inequalities as equalities using slack variables. Although with equalities only we lose the ability to explicitly encode non-negativity constraints, we mitigate this by regularization making sure that solutions remain bounded.

### A.4  EFFICIENT FORWARD AND BACKWARD COMPUTATIONS

**Forward propagation and solving the quadratic program.** We can solve the quadratic program directly with well-known techniques (Wright and Nocedal, 1999), namely by simplifying and solving the following KKT system for some $\lambda \in \mathbb{R}^m$,

$$\begin{bmatrix} G & A^\top \\ A & 0 \end{bmatrix} \begin{bmatrix} -z \\ \lambda \end{bmatrix} = \begin{bmatrix} \delta \\ -\beta \end{bmatrix} \tag{23}$$

For smaller problems, we can solve this system efficiently using a dense Cholesky factorization. For larger problems, we use an indirect conjugate gradient method to solve the KKT system using only *sparse* matrix computations. Both methods are performed batch parallel on GPU.

**Backward propagation and gradients computation.**  In the backward pass, we need to update the ODE coefficients in the constraint matrix $A$ and $\beta$. We obtain the gradient relative to constraint matrix $A$ by computing $\nabla_A \ell(A) = \nabla_A z \nabla_z \ell(z)$, where $\ell(.)$ is our loss function and $z$ is a solution of the quadratic program.

We can compute the individual gradients using already established techniques for differentiable optimization Amos and Kolter (2017) with the addition of computing sparse gradients only for the constraint matrix $A$. Briefly, computing the gradient requires solving the system equation 23 for with a right-hand side containing the incoming gradient $g$:

$$-\begin{bmatrix} G & A^\top \\ A & 0 \end{bmatrix} \begin{bmatrix} d_z \\ d_\lambda \end{bmatrix} = \begin{bmatrix} g \\ 0 \end{bmatrix}. \tag{24}$$

The gradient $\nabla_A \ell(A)$ can then be computed by first solving for $d_z, d_\lambda$ and then computing $d_\lambda z^\top + \lambda d_z^\top$ (Amos and Kolter, 2017). In general, this would produce a dense gradient matrix, which is very memory inefficient for sparse $A$. We avoid this by computing gradients only for the non-zero terms of $A$ by computing sparse outer products.

## A.5 CENTRAL DIFFERENCE FOR HIGHEST ORDER

The method proposed in Young (1961) does not add a smoothness constraint for the highest order derivative term, since derivative information is not available. In cases where a more accurate highest order term is required, we also add a central difference constraint as a smoothness condition on the highest order term.

## A.6 NONLINEAR ODES

The standalone solver described above works for linear ODEs. When combined with neural networks, we can extend the approach to nonlinear ODEs by combining solving with learning.

For each non-linear ODEs term $g_k$, we add an extra variable $\nu_{k,t}$ with coefficients $\phi_{k,t,x}$ corresponding to the non-linear term to our linear program.

We rewrite our nonlinear ODE in equation 4 as

$$\sum_{i=0}^{d} c_{i,t,x} u_t^{(i)} + \sum_{k=0}^{r} \phi_{k,t,x} \nu_{k,t} = b_{t,x} \tag{25}$$

$$\nu_{k,t} = g_k(t, u_t, u_t', \ldots), \, k = 0, ..., r \tag{26}$$

$$\text{s.t. } [u_{t=1}, u_{t=1}', ...] = \omega_x, \tag{27}$$

that is, for each nonlinear term $k$ and for every time step $t$ we also add in $z$ an auxiliary variable $\nu_{k,t}$. Additionally, we include derivative variables $\nu_{k,t}^i$ that are part of the Taylor approximations to ensure smoothness. We then solve the linear part of the above ODE, that is equation 25 subject to 27 with the linear programs we described in the previous subsections. Further, we convert the nonlinear part in equation 26 to a loss term $\left(\nu_{k,t} - g_k(t, u_t, u_t', \ldots)\right)^2$, which is added to the loss function of the neural network. With the extra losses, we learn the parameters $\phi$ such that $\nu_k$ is close to the required non-linear function of the solution.

Figure 14 shows solving and fitting of a non-linear ODE.

**Nonlinear ODEs for Discovery.** When building MNN models for governing equation discovery, we incorporate nonlinear ODEs using a set of predefined basis functions $\{\theta_i\}$, such as the polynomial basis functions (Brunton et al., 2016), to build an equation of the form

$$\frac{d}{dt} u(t; x) = \sum_i^k \theta_i(u_x'(t)) \tag{28}$$

The input $u_x'(t)$ to the basis functions are generated by a neural network with input $x$ as $u_x' = \text{NN}(x)$, where $u_x'$ (and possibly $x$) depends on time $t$. To ensure that this is a proper nonlinear ODE we add a consistency term to the loss function to minimize the squared loss $(u(t; x) - u_x'(t))^2$. This ensures that the basis input and ODE solution are close.

## A.7 COMPLEXITY

The computational and memory complexity of MNNs is determined by the size of the time grid $n$, and the order $d$ of the ODEs to be generated. The last layer of $f$ outputs $n \times (d + 2)$ ODE parameters. This means that the memory required to store the coefficients can be large depending on the grid size and dimension. The main computational effort in solving the system equation 23 for a batch of ODEs, which we do by a Cholesky factorization for small problems or sparse conjugate gradient for large ones. Cholesky factorization has complexity cubic in $nd$ while conjugate gradient has quadratic complexity.

## A.8 NUMERICAL VALIDATION OF THE SOLVER

**Benchmarking against RK4 from `scipy` and `torchdiffeq`.** We compare with traditional ODE solvers on second- and third-order linear ODEs with constant coefficients from the `scipy` package. For a *learning* comparison, we also compare with the RK4 solver with the adjoint method from the `torchdiffeq` package on the benchmark task of fitting long and noisy sinusoidal functions of varying lengths. The quantitative and qualitative results in Appendix D show that NeuRLP

is comparable to standard solvers on the linear ODE-solving task. On the *fitting* task NeurLP significantly improves upon the baseline and is *about 200x faster* with a lower MSE loss than the `torchdiffeq` baseline for 1000 steps.

**NeuRLP can learn time steps.** Unlike traditional solvers, NeuRLP can learn the discretization grid for learning and solving ODEs, becoming adaptively finer in regions where the fit is poor. We validate this on fitting a damped sinusoid, see results in Figure 13, where we begin with a uniform grid and with steps becoming denser in regions with bad fit.

## B    FURTHER DETAILS FOR SECTION 4

### B.1    DISCOVERY OF PHYSICAL PARAMETERS

**Problem.** Often, the problem is not to discover the governing equations in a system but the most fitting physical parameters explaining the observations. Applications include inverse problems in dynamical systems (Wenk et al., 2020).

**Gold standard.** We use second order Neural ODEs (Norcliffe et al., 2020) to fit ODE models of corresponding to Newton's second Law, matching corresponding derivative coefficients to infer the physical parameters.

**Mechanistic NNs for discovering physical parameters.**
We use a second order ODE MNN with a time invariant 2nd order coefficient to match Newton's second Law. The force is learned by a neural networks as a function of position.

**Experiment.** We design an experiment with two bodies with masses $m_1 = 10, m_2 = 20$, distance $d$ and initial velocities $v_1, v_2$, moving under the influence of Newtonian laws, and gravitational force, $F = G\frac{m_1 m_2}{r^2}\hat{r}$, $\hat{r}$ being the unit vector of direction of force, $G = 2$ the gravitational constant. We generate a single random train trajectory for the two bodies for 40k steps. The physical parameters we infer are mass ratio $\frac{m_1}{m_2}$ and distribution of force values $F = [F_x, F_y]$, by combining Newton's second and third law. We show quantitative and qualitative results in Figure 5. Since forces are only determined up to a constant, to compare forces we normalize by dividing by the force at the first step. Neural ODE and Mechanistic NNs estimate the mass ratio while MNNs perform significantly better at estimating the force distribution and Neural ODE forces often have the incorrect direction as shown by the negative cosine similarity averaged over the entire trajectory.

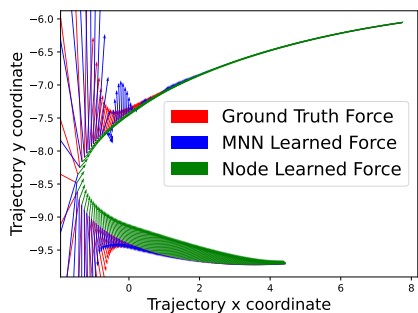

| Method | Force MSE ↓ | Cosine sim. ↑ | Mass Ratio GT=2 |
|--------|-------------|---------------|-----------------|
| SONODE | 879 | -0.26 | 2.11 |
| MNN | 345 | 0.85 | 2.02 |

Figure 5: Normalized true and learned force vectors during 550 steps for 2-body parameter discovery and comparison.

### B.2    FORECASTING FOR TIME SERIES

**Problem.** Time-series modelling and future forecasting is a classical statistical and learning problem, usually with low-dimensional signals, like financial data or complex dynamical phenomena from sciences.

**Gold standard.** We compare with Neural ODE and variants including second order and augmented Neural ODEs.

**Mechanistic NNs for time series.** We use a basic Mechanistic NN second-order ODE for this experiment.

**Experiment.** We validate on the benchmark of modeling the accelerations $a_2$ produced over time by a shaker under a wing in aircrafts (Norcliffe et al., 2020). The model sees 1,000 past time accelerations $a_2$ and predicts the next 4000. We show quantitative results in and qualitative

results in Figure 10 and C.3 in the appendix. The distribution of predicted $a_2$ at test time are very close to the true ones. Mechanistic NNs are on-par with second-order ODE, converge significantly faster, and achieve two times lower training error showing they can model complex phenomena.

### B.3 ODE DISCOVERY

We give further details regarding the ODE discovery setup.

This method follows the SINDy Brunton et al. (2016) approach for discovering sparse differential equations using a library of basis functions. Unlike SINDy, which resorts to linear regression, the MNN method uses deep neural networks and builds a non-linear model which allows modeling of a greater class of ODEs.

The method requires a set of basis functions such as the polynomial basis functions up to some maximum degree. Over two variables $x, y$ this is the set of functions $\{0, x, y, x^2, xy, y^2, xy^2, \ldots, y^d\}$ for some maximum degree $d$. Let $k$ denote the total number of basis functions.

Next we are given some observations $X = [(x_0, y_0), (x_1, y_1), \ldots, (x_{n-1}, y_{n-1})]$ for $n$ steps. We first transform the sequence by applying an MLP to the flattened observations producing another sequence of the same shape.

$$\tilde{X} = [(\tilde{x}_0, \tilde{y}_0), \ldots, (\tilde{x}_{n-1}, \tilde{y}_{n-1})] = \text{MLP}(X)$$

We apply the basis functions to $\tilde{X}$ to build the basis matrix $\Theta \in \mathbb{R}^{n \times k}$.

$$\Theta(\tilde{X}) = \begin{bmatrix} 1 & \tilde{x}_0 & \tilde{y}_0 & \tilde{x}_0^2 & \tilde{x}_0 \tilde{y}_0 & \tilde{y}_0^2 & \cdots \\ 1 & \tilde{x}_1 & \tilde{y}_1 & \tilde{x}_1^2 & \tilde{x}_1 \tilde{y}_1 & \tilde{y}_1^2 & \cdots \\ \vdots & \vdots & \vdots & \vdots & \vdots & \vdots & \\ 1 & \tilde{x}_{n-1} & \tilde{y}_{n-1} & \tilde{x}_{n-1}^2 & \tilde{x}_{n-1} \tilde{y}_{n-1} & \tilde{y}_{n-1}^2 & \cdots \end{bmatrix} \tag{29}$$

Let $\xi \in \mathbb{R}^{n \times 2}$ be a set of parameters, with each column specifying the active basis functions for the corresponding variable in $[\dot{x}, \dot{y}]$.

The ODE to be discovered is then modeled as

$$[\dot{x}, \dot{y}] = f(\Theta(\tilde{X})\xi) \tag{30}$$

where $f$ is some arbitrary differentiable function. Note that for SINDy $\tilde{X} = X$ and $f$ is the identity function and the problem is reduced to a form of linear regression adapted to promote sparsity in $\xi$. SINDy estimates the derivatives using finite differences with some smoothing methods.

With MNN the ODE 30 is solved using the quadratic programming ODE solver to obtain the solution $\bar{x}_t, \bar{y}_t$ for $t \in \{0, \ldots, n-1\}$. The loss is then computed as the MSE loss between $\tilde{x}_t, \tilde{y}_t, \bar{x}_t, \bar{y}_t$ and the data $x_t, y_t$.

$$\text{loss} = \frac{1}{N} \sum_t (\tilde{x}_t - x_t)^2 + (\tilde{y}_t - y_t)^2 + (\bar{x}_t - x_t)^2 + (\bar{y}_t - y_t)^2$$

We show two examples of cases where $F(u)$ is a rational function (a ratio of polynomials) and when $F(u)$ is a nonlinear function of $\Theta\xi$. Moreover, unlike SINDy, MNN can learn a single governing equation from multiple trajectories each with a different initial state making MNN more flexible. In many situations a single trajectory sample is not enough represent to the entire state space while multiple trajectories allow discovery of a more representative solution.

**Planar and Lorenz System.** We first examine the ability of MNN equation discovery for systems where the true ODE can be exactly represented as a linear combination of polynomial basis functions. We use a two variable planar system and the chaotic Lorenz system as examples. Both MNN and SINDy are able to recover the planar system. Simulation of the learned Lorenz ODE are shown in Figure 6 for MNN and SINDy.

Next we consider ODE systems where the derivative *cannot* be written as a linear combination of polynomial (or other) basis function.

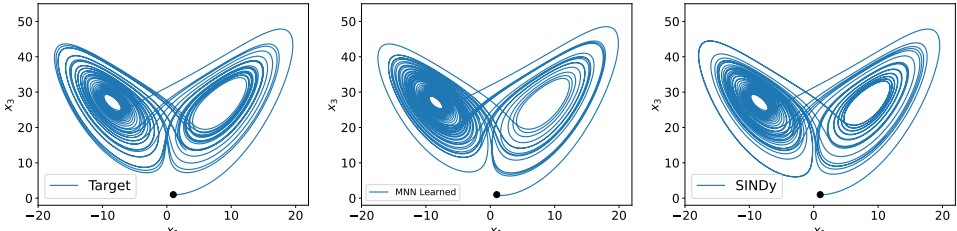

Figure 6: Learned ODEs for the chaotic Lorenz system. Showing the true trajectory, the MNN learned ODE trajectory and SINDy learned ODE trajectory.

**Nonlinear Function of Basis.** First, we consider systems where the derivative is given by a nonlinear function of a polynomial. For simplicity we assume that the nonlinear function is known. As an example we solve the system from Figure 2 with the tanh nonlinear function.

Vector fields for the learned systems are shown in Figure 2 for MNN and SINDy. We see that although SINDy fits the training example, the directions diverge further away. With MNN we see that the learned vector field is consistent with the ground truth far from the training example even though we use only a single trajectory.

**Rational Function Derivatives.** Second, we consider the case where the derivative is given by a rational function, i.e., $F(u) = p(u)/q(u)$, where $p$ and $q$ are polynomials. Such functions cannot be represented by the linear combination of polynomials considered by standard SINDy, however such functions can be represented by MNNs by taking $p$ and $q$ to be two separate combinations of basis polynomials and dividing. An example is shown in Figure 7 in the appendix for the system where we see again MNNs learning much better equations compared to SINDy with a second-order polynomial basis tha overfits. Further, by including more trajectories in the training, results improve further, see Figure 7.

## C    EXPERIMENTAL DETAILS

### C.1    DISCOVERY OF GOVERNING EQUATIONS

#### C.1.1    DISCOVERED EQUATIONS.

**MNN Lorenz**

$$x' = -10.0003x + 10.0003y$$
$$y' = 27.9760x + -0.9934y - 0.9996xz$$
$$z' = -2.6660z + 0.9995xy$$

**SINDy Lorenz**

$$x' = -10.000x + 10.000y$$
$$y' = 27.998x + -1.000y + -1.000xz$$
$$z' = -2.667z + 1.000xy$$

**MNN Non-linear**

$$x' = \tanh(-0.7314x + 0.5545y +$$
$$- 1.2524x^2 + -0.1511xy + 0.2134y^2)$$
$$y' = \tanh(0.9879x + 1.0005y + 0.1742x^2)$$

**SINDy Non-linear**

$$x' = -1.968x + 0.985y + -0.054x^2$$
$$y' = 1.466y + 11.892x^2 + -5.994xy + 0.085y^2$$

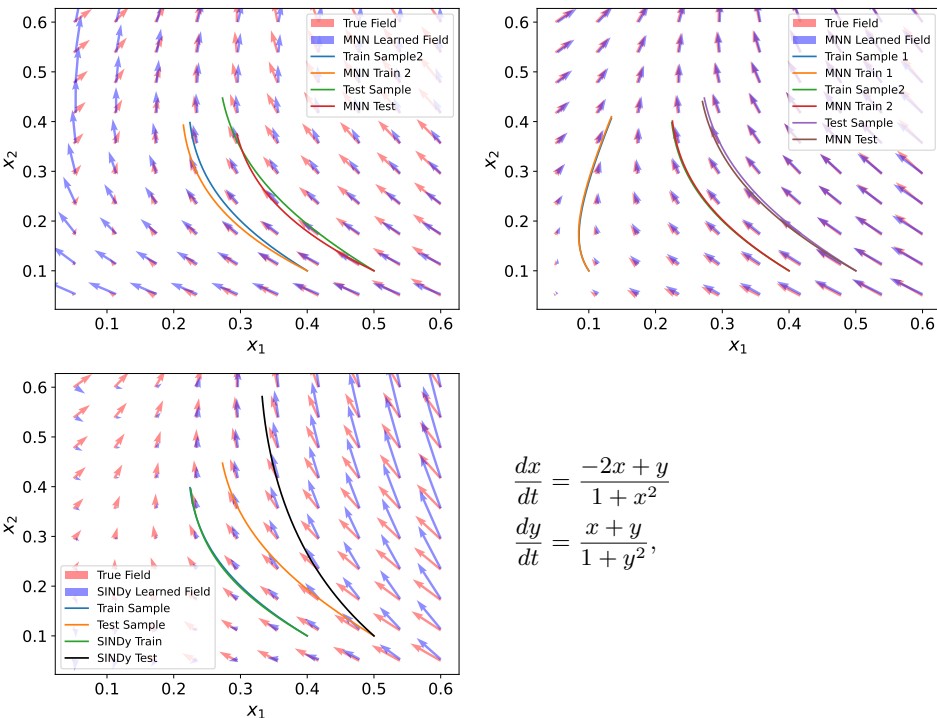

Figure 7: Learned ODE vector fields for MNN and SINDy with rational function derivatives and one and two training trajectories. MNN can handle multiple input examples. The ground truth ODE is also shown.

$$\frac{dx}{dt} = \frac{-2x + y}{1 + x^2}$$
$$\frac{dy}{dt} = \frac{x + y}{1 + y^2},$$

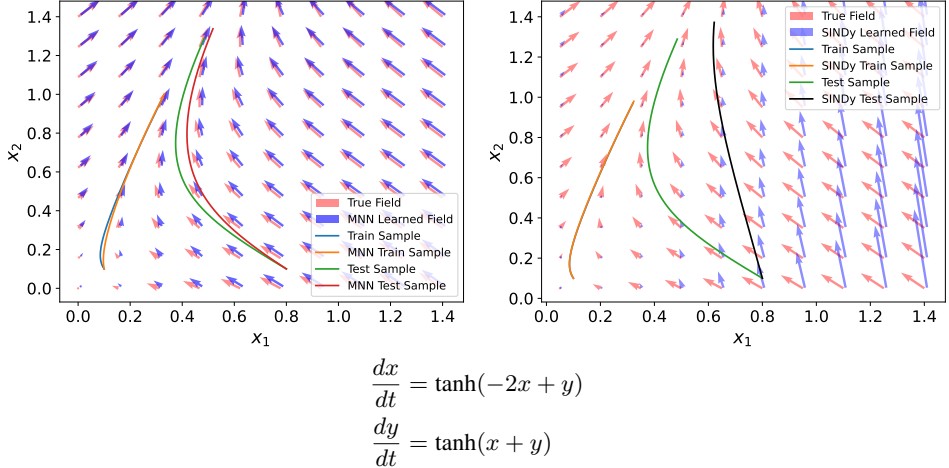

$$\frac{dx}{dt} = \tanh(-2x + y)$$
$$\frac{dy}{dt} = \tanh(x + y)$$

Figure 8: Learned ODE vector fields for Mechanistic NN and SINDy with non-linear tanh function of basis combination and training and test trajectories.

**MNN Rational**

$$x' = \frac{-0.9287x + 0.4386y + -1.1681x^2 + 0.3545y^2}{0.4871 + 0.8123x + 0.0984x^2 + 0.3700xy + 0.3081x^2}$$
$$y' = \frac{0.6360x + 0.5971y + 0.3267x^2}{0.6090 + 0.7507x^2 + 0.5694y^2}$$

**SINDy Rational**

$$x' = -1.705x + 0.899y + -0.318x^2$$
$$y' = -0.795 + 3.072y + 4.777x^2 + 6.892xy + -4.681y^2$$

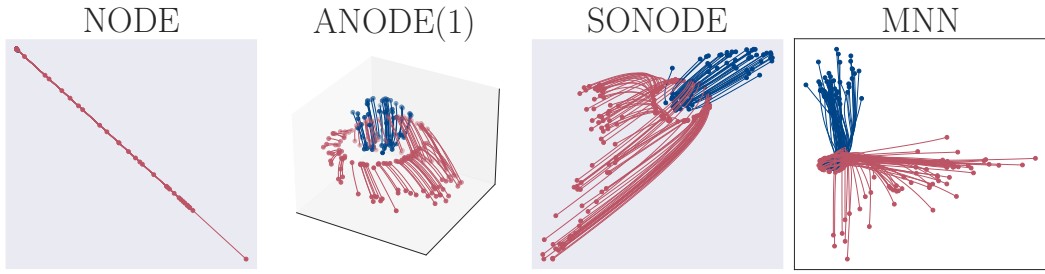

Figure 9: (a) Visualizing the state evolution of the learned equations $\mathcal{U}_x$ data points in nested spheres. The points from the two classes are perfectly separated despite the nested topology without requiring augmentations.

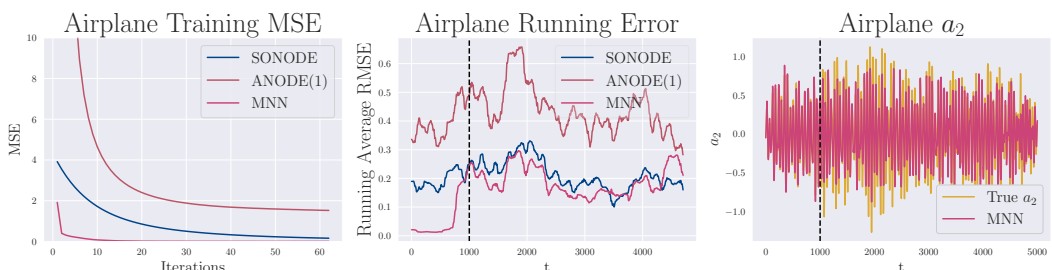

Figure 10: Modeling airplane vibrations.

## C.2 Nested Spheres

We test MNN on the nested spheres dataset (Dupont et al., 2019), where we must classify each particle as one of two classes. This task is not possible for unaugmented Neural ODEs since they are limited to differomorphisms (Dupont et al., 2019). We show the results in Figure 9, including comparisons with Neural ODE (Chen et al., 2018), Augmented Neural ODE and second-order Neural ODE (Norcliffe et al., 2020). MNNs can comfortably classify the dataset without augmentation and can also derive a governing equation.

We use a second order ODE with coefficients computed with a single layer and the right hand side is set to 0. We use a step size of 0.1 and length 30. However, as we note, 5 time steps are enough for accurate classification. The loss function is the cross entropy loss.

MNNs obtain an explicit linear ODE per datapoint that governs the evolution of the point. The example we give is for one of the ODEs for one point and for a 5-time step evolution. This computed equation is sufficient for perfect classification.

## C.3 Airplane Vibrations

MNNs can learn complex dynamical phenomena significantly faster than Neural ODE and second order Neural ODE. We reproduce an experiment with a real-world aircraft benchmark dataset (Noël and Schoukens, 2017; Norcliffe et al., 2020). In this dataset the effect of a shaker producing acceleration under a wing gives rise to acceleration $a_2$ on another point. The task is to model acceleration $a_2$ as a function of time using the first 1000 step as training only and to predicting the next 4000 steps. Results of the experiment are shown in Figure 10. We compare against Augmented Neural ODE and second order Neural ODE. MNNs are on-par with second-order ODE, converge significantly faster in the number of training steps, and achieve two times lower training error, showcasing the capacity for modeling complex phenomena and improving with modest architectural modifications. The predicted $a_2$ accelerations are very close to the true ones in the center-right plot.

| Method | RMSE |
|---|---|
| NNLi et al. (2020c) | 0.1716 |
| FCNLi et al. (2020c) | 0.0253 |
| PCANN Bhattacharya et al. (2021) | 0.0299 |
| RBMLi et al. (2020c) | 0.0244 |
| GNO Li et al. (2020a) | 0.0346 |
| LNO Li et al. (2020c) | 0.0520 |
| MGNOLi et al. (2020b) | 0.0416 |
| FNO Li et al. (2020c) | 0.0070 |
| Mechanistic NN | 0.0065 |

Table 2: PDE results on 2d Darcy flow

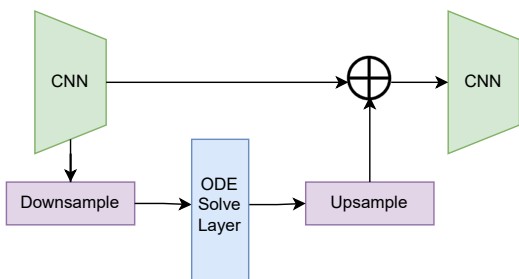

Figure 11: PDE module architecture used for 2d data

For this experiment (Section B.2) we use an MNN with a second order ODE, step size of 0.1 and 200 steps during training. The coefficients and constant terms are computed with MLPs with 1024 hidden units.

### C.4 Discovering Mass and Force Parameters.

For this part of the experiment we use an MNN with a restricted ODE to match Newton's second law. In the MNN model for this experiment, we use the same coefficient for the second derivative term for all time steps with the remaining coefficients fixed to 0, that is $c_2(t) = c$ and $c_1(t) = 0, c_0(t) = 0$. $b(t) = F_t$ corresponds to the force term which is computed by a neural network from the initial position and velocity with two hidden layers of 1024 units and Newton's second law $F_{21} = -F_{12}$. We use a step size of $0.01$ and run for 50 epochs.

The baseline is an SONODE designed to correspond to Newton's second and third law with an MLP for force as above.

### C.5 PDE Solving

**1d Model.** For 1d problems we use a simplest possible model of modeling the spatial dimension by independent ODEs. We use a history of 10 time steps and predict for 9 time steps in one iteration, using the last time step as initial condition for the ODE. During evaluation we predict and evaluate for 100 steps. We use 3rd and 4th order ODEs. The coefficients for ODEs, step sizes the right hand side ($b$) are computed by 1d ResNets with 10 blocks. We use the L1 loss which we find improves rollout performance.

**2d Model.** In Figure 11 we show the MNN architecture we used to solve PDEs. We use the 2d Darcy Flow dataset used by Li et al. (2020c) scaled to 85x85. The ODE is solved for 30 steps and the entire soluton trajectory is then upsampled and combined with the input features map. The network is built by stacking three such modules together plus an input MLP layer and an output layer.

## D    FURTHER EXPERIMENTS

### D.1    VALIDATING THE NEURLP ODE SOLVER

First we examine whether our quadratic programming solver is able to solve linear ODEs accurately. For simplicity we choose the following second and third order linear ODEs with constant coefficients.

$$u'' + u = 0 \tag{31}$$

$$u''' + u'' + u' = 0 \tag{32}$$

For the NeuRLP solver we discretize the time axis into 100 steps with a step size of 0.1. We compare against the ODE solver `odeint` included with the SciPy library. The results are shown in 12 where we show the solutions, $u(t)$, for the two ODEs along with the first and second derivatives, $u'(t), u''(t)$. The results from the two solvers are almost identical validating the quadratic programming solver.

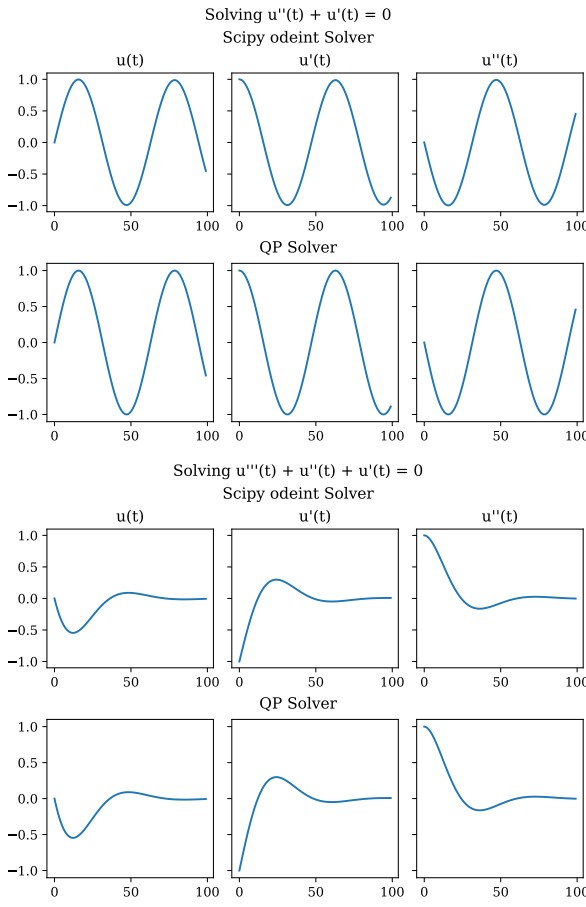

Figure 12: Comparing ODE solvers on 2nd and 3rd order ODEs.

Next we examine the ability of the solver to learn the discretization. We learn an ODE to model a damped sine wave where each step size is a learanable parameter initial to 0.1 and modeled as a sigmoid function. We show the results in Figure 13 for a sample of training steps. We see the step sizes varying with training and the steps generally clustered together in regions with poorer fit.

Next we demonstrate a non-linear equation. For this we introduce a variable in the QP solver for a non-linear term add a squared loss term as described in the paper. We use the equation $c_2(t)y'' + c_1(t)y' + c_0(t)y + \phi(t)y^2 = 1$, with time varying coefficients and fit a sine wave. The result is in Figure 14. The ODE fits the sine wave and at the same time the non-linear solver term fits the true non-linear function of the solution.

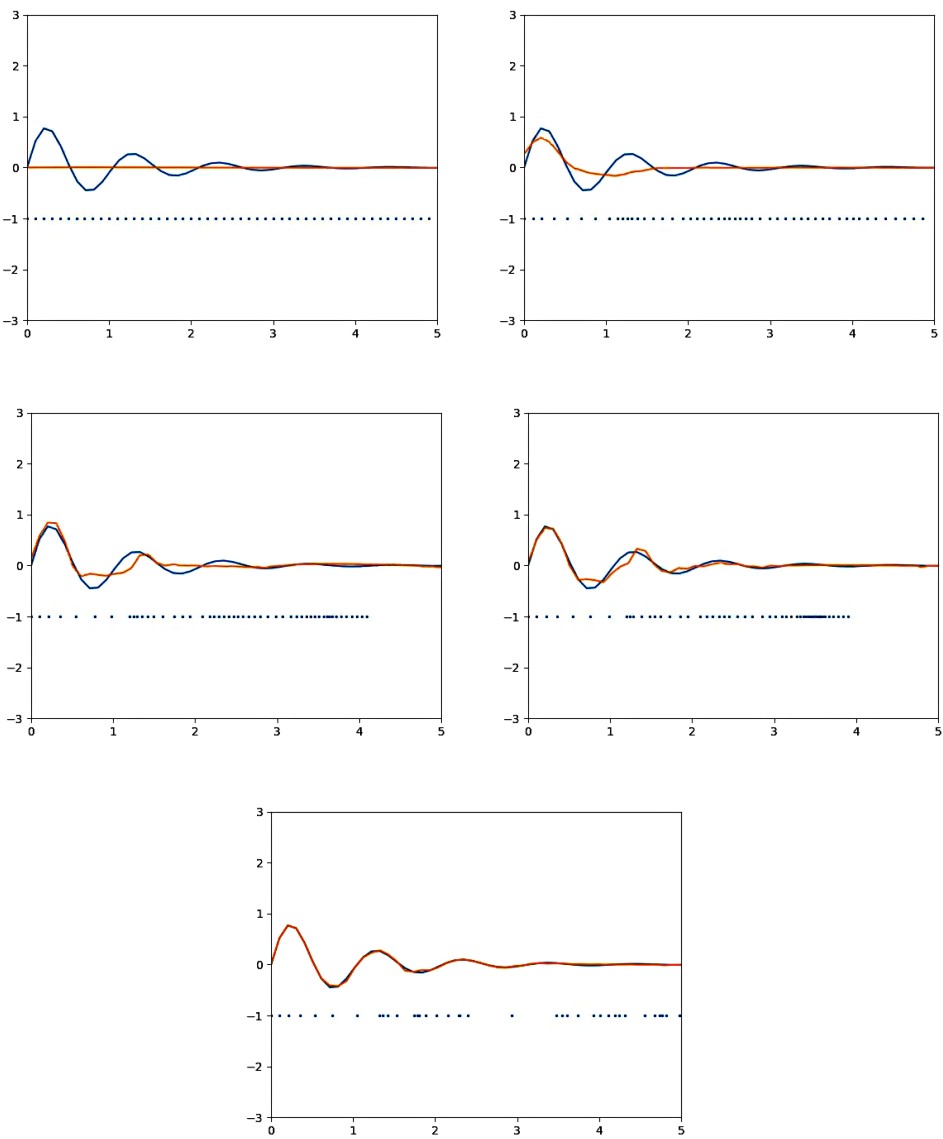

Figure 13: Demonstrating a learned grid for fitting a damped sinuoidal wave (blue curve) over the course of training. The dots show the learned grid positions. The grid generally becomes finer for regions where the fit is poorer.

## D.2 LEARNING WITH NOISY DATA

We perform an simple experiment illustrate how the ODE learning method can fit ODEs to noisy data. We generate a sine wave with dynamic Gaussian noise added during each training step. We train two models: the first a homogeneous second order ODE with arbitrary coefficients and the second a homogeneous second order ODE with constant coefficients. We also train a model without noise. The results are shown in Figure 15. The figures show that the method can learn an ODE in the presence of noise giving a smooth solution. The model with constant coefficients learns the following ODE.

$$0.92023u'' - 0.00016u' + 0.228u = 0,$$

with (learned) initial conditions $u(0) = -0.031799$ and $u'(0) = 2.3657$.

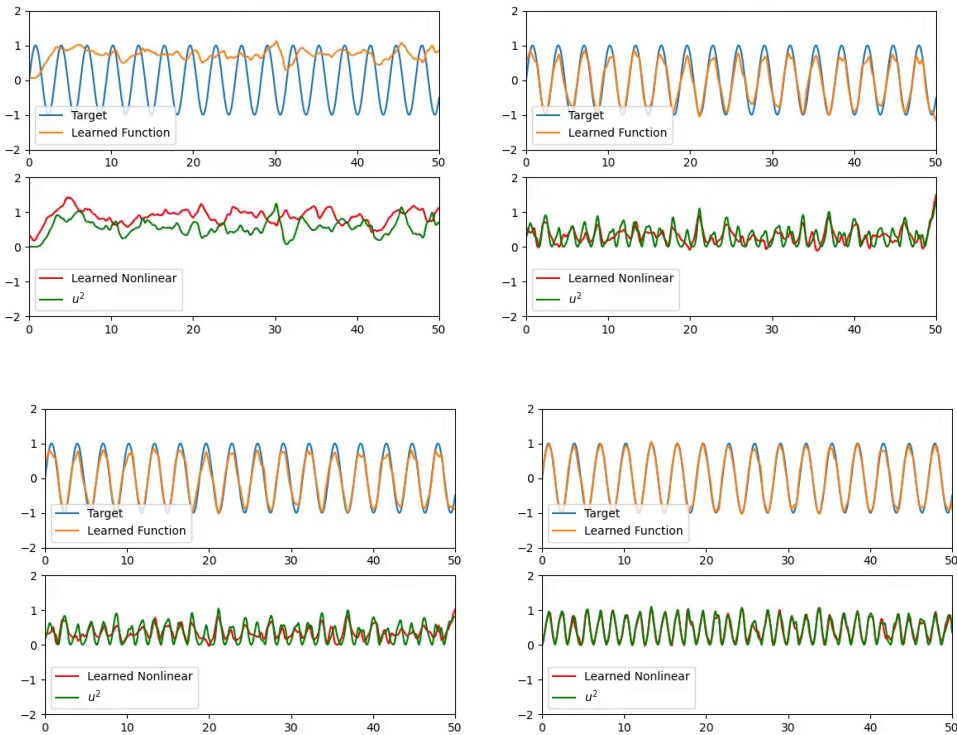

Figure 14: Demonstrating fitting a sine wave with a non-linear ODE $c_2(t)y'' + c_1(t)y' + c_0(t)y + \phi(t)y^2 = 1$. The non-linear function is $y^2$ and the bottom shows the solver variable fitting the non-linear function.

### D.3    COMPARING RK4 WITH THE NEURLP SOLVER

We compare NeuRLP with the RK4 solver from *torchdiffeq* on a task of fitting noisy sinusoidal waves of varying lengths. We compare MSE and time in Table 3 and Figure 16.

Table 3: Comparing the NeuRLP solver with the RK4 solver with a step size of 0.1 on fitting noisy sinusoidal waves of 300 and 1000 steps. Showing MSE loss and time.

| Steps | QP (seconds) | RK4 (seconds) | QP Loss | RK4 Loss |
|-------|--------------|---------------|---------|----------|
| 40    | 1.52         | 28.06         | 11.4    | 29.3     |
| 100   | 1.61         | 64.57         | 27.9    | 35.6     |
| 300   | 1.76         | 211.52        | 52      | 96.8     |
| 500   | 2.12         | 359.7         | 128     | 301      |
| 1000  | 3.68         | 666.69        | 292     | 589      |

### D.4    2-BODY PROBLEM

We show learned trajectories for a 2-body prediction problem with an MNN on synthetic data in Figure 18. The objects are generated using the gravitation force law for 4000 steps and the first half are used for training and we predict the second half.

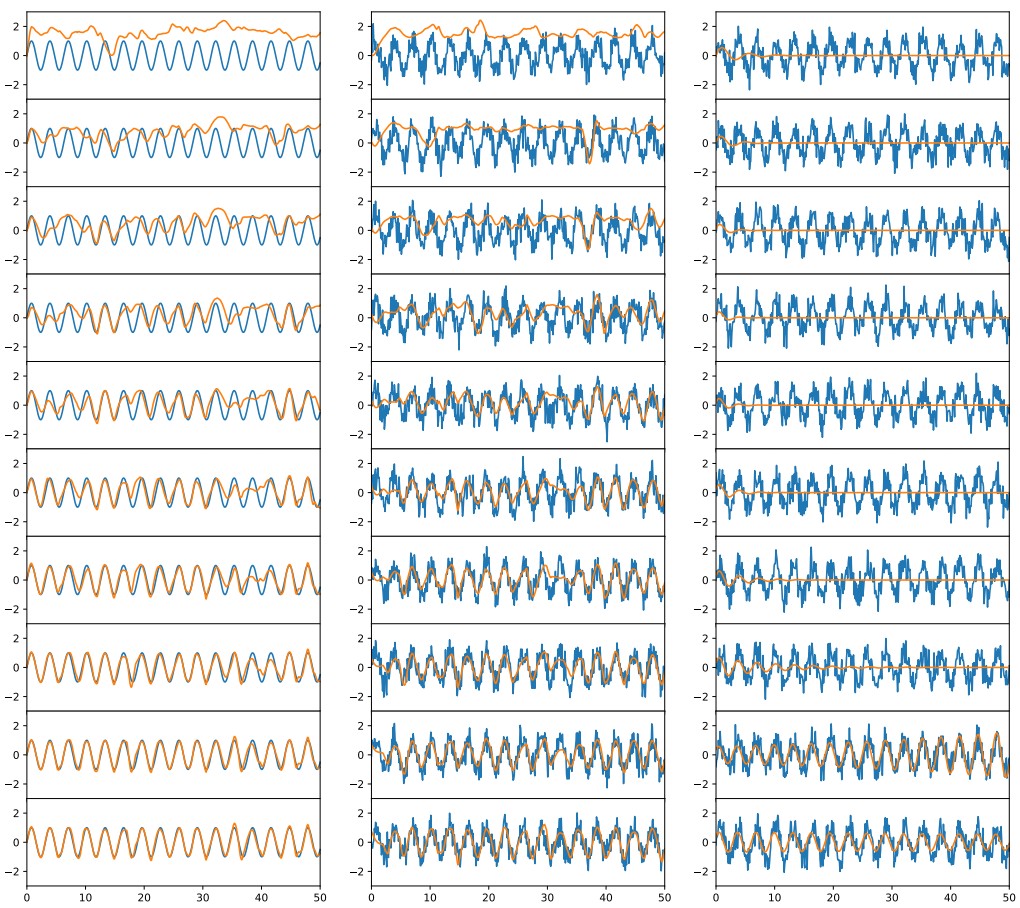

Figure 15: Learning sine waves without and with dynamically added Gaussian noise with 2nd order ODE with arbitrary coefficients (middle) and constant coefficients (right). The figure on the right corresponds to the ODE $0.92023u'' - 0.00016u' + 0.228u = 0$.

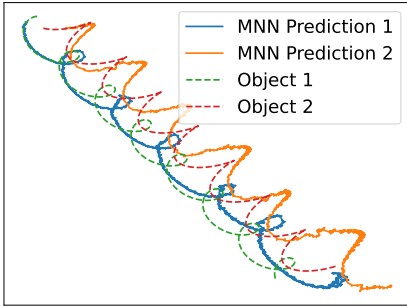

Figure 18: 2-body problem: Predicted orbits for MNN

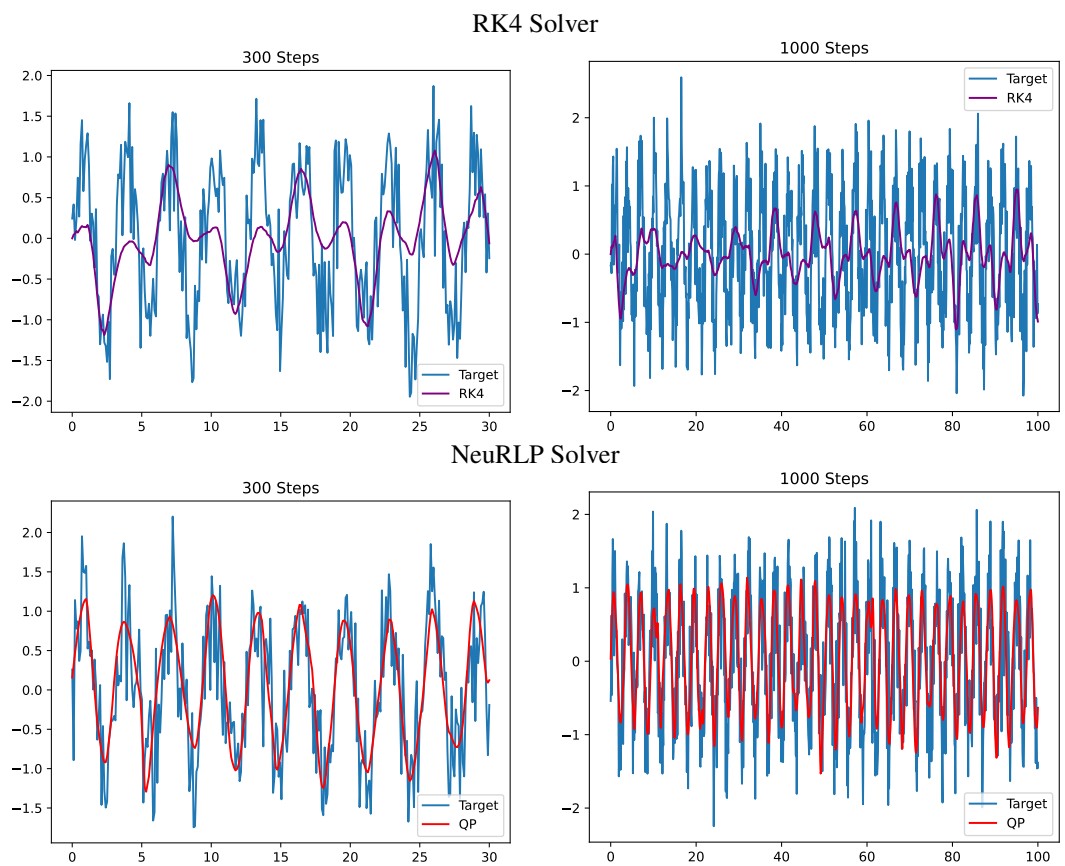

Figure 16: Comparison of RK4 solver from *torchdiffeq* and our NeuRLP solver for fitting sinusoidal waves with Gaussian noise added at each iteration. Length of the wave and number of steps is 300 (left column) and 1000 (right column). Step size is 0.1. Trained for 100 iterations. The NeuRLP solver has better performance (and efficiency) for longer trajectories.

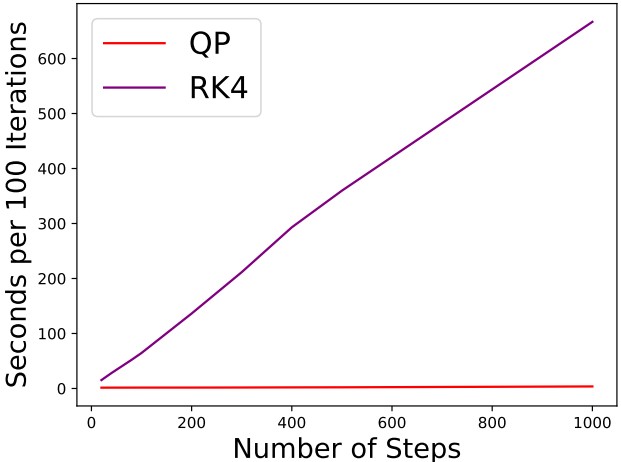

Figure 17: Number of seconds per 100 iterations for fitting noisy sinusoidal waves. The NeuRLP solver is significantly more efficient over longer times due to its parallelism.

