# OpenReview forum: "Mechanistic Neural Networks for Scientific Machine Learning"
_ICLR.cc/2024/Workshop/AI4DiffEqtnsInSci — AI4DiffEqtnsInSci @ ICLR 2024 Poster_

### Official Review · Reviewer_tPgU · 2024-02-29
**Reasonable and exhaustive experimental results with several unclear aspects**

**Rating:** 5
**Confidence:** 4

**Review:**

### Summary:
Mechanistic Neural Networks (MNN) are introduced for equation discovery, PDE solving, and forecasting. The method implements a mechanistic encoder that translates the input into an ODE and, at the same time, predicts the step size to solve the equation. Subsequently, a neural Relaxed Linear Programming Solver integrates the inputs to generate a prediction. Results underline MNN's ability to discover equations and accurately forecast dynamic systems, while not qyite reaching the accuracy of state-of-the-art methods in PDE solving.

### Strengths:
- Exhaustive experimentation with additional details provided in the appendix.
- Wide applicability of the method to different tasks.
- Accelerating equation discovery by providing an efficient GPU implementation for parallel computing.

### Weaknesses and Concerns
- Unclear what the exact inputs and outputs of the model components are. E.g., what does $f_{\theta}$ receive and concretely produce as output and how are these outputs then composed as an ODE?
- MNN is compared against a fairly old version of SINDy. More recent variants, such as SINDy-PI [[1]](https://royalsocietypublishing.org/doi/full/10.1098/rspa.2020.0279) in fact are able to recover the coefficients of nonlinear systems.
- While MNNs can be applied to various tasks, it does not seem to reach FNO's performance in PDE solving.
- In Figure 2, it is unclear to me how MNN can technically learn the entire vector field accurately (including the region of the test sample) by only seeing the train sample. The SINDy solution looks realistic to me, given the method only receives the plotted training trajectory in the left end of the vector field. I understand that the two trajectories plotted in Figure 8 are more informative about the shape of the vector field, allowing MNN to better capture the overall vector field in that case, i.e., when receiving two trajectories. Can you explain why Figure 2 only contains one trajectory, verify whether MNN is only receiving that single trajectory, and explain how MMN technically can get such an accurate notion of the complete vector field then? Also, how does SINDy (or SINDy-PI) perform when receiving two trajectories during training?

### Questions:
1. On page 1, what are large batches and do you seek to learn many ODEs simultaneously from different families, or do you vary some coefficients in an ODE and call this many ODEs?
2. If the step size is a learnable parameter, how do you ensure that it does not converge to zero in order to generate the most accurate results?
3. Can you show some examples of what $\omega_x$ converges to? Would be interesting to see what step size it induces under what circumstances.
4. In Equation (2), can you specify what $b$ is and why the right-hand-side is not zero as in traditional ODE representations? That is, do you allow for a bias depending on $t$ and $x$ and how do you account for it and model it?
5. in "Efficient Quadratic relaxation and extension", is the LP method standing for linear programming? Maybe add as explanation.

### Minor comments:
- On page 1, space missing in "When training ODE representationswe..."
- In the \textbf{} commands, when opening new paragraphs, you sometimes do and do not use a dot, which might be made consistent.
- At the beginning of Section 4, PDE solving printed two times in "... discovery of equations, PDE solving, , PDE solving ..."
- Typo in **N-body prediciton** on page 4.

### Assessment
In balance I am undecided about this paper, which, on the one hand, offers exhaustive and reasonable results on a relevant topic, but, on the other hand, is vague about several aspects that need clarification.

### References:
[1] https://royalsocietypublishing.org/doi/full/10.1098/rspa.2020.0279

---

### Official Review · Reviewer_kRyg · 2024-03-01
**Proposed NeuRLP solver is novel and important addition for SciML; Experiments with proposed architecture need more clarifications**

**Rating:** 8
**Confidence:** 4

**Review:**

Paper proposes NeuRLP solver that uses differentiable quadratic programming to solve ODEs as an alternative to traditional solvers. The solver is used within the proposed Mechanistic NN block that consists of an ODE representation whose parameters are learned with an neural encoder. Output of the MNN block is obtained by solving the corresponding ODE.


Strengths:
1. NeuRLP uses differentiable quadratic programming with equality constraints to solve the ODE. Method can be extensively parallelized, even across time steps. Table 2 and Figure 18 show that proposed method is significantly faster than the traditional solvers like RK4, especially on longer sequences. This component can be beneficial in many SciML methods that incorporate ODE within the architecture.
2. NeuRLP is able to solve ODEs with nonlinear terms using auxiliary variables in the quadratic programming problem.
3. Experiments using NeurLP and MNN show improvements in forecasting tasks in ODEs and PDEs.

Weaknesses:
1. Section 2 can be clarified to better distinguish the "ODE discovery" and "Prediction/forecasting" cases of the MNN models. Currently, these cases seem disconnected and require a back-and-forth with Appendix.
2. Nonlinear ODE discovery experiments: I am not convinced if the proposed MNN method is better than SINDy for discovering ODEs.
	1. In Appx B.3, MLP(X) is used to obtained a nonlinear representation $\tilde{X}$ of X and the ODE is learnt over this representation. But the loss function seems to enforce that $\tilde{X} = X$ which negates the use of this MLP. This can also be seen from the learnt equations in C.1 that do not contain the effect of MLP.
	2. When the ground truth dynamics has a nonlinear function applied at **last**, e.g., tanh or of the form p/q, MNN is given the exact form of this function whereas SINDy is not. I believe it should be easy to adapt SINDy to the case when this final function is known and nonlinear.
3. Since the paper claims interpretability, it would be good to show the equations learnt within MNN for all experiments (e.g., n-body experiments).
4. Please provide the following details in the paper:
    1. What was the exact family of ODEs $U_x$ used in experiments (e.g., what were $g_k$)?
    2. What was the maximum number of ODE terms used in experiments? Do the proposed methods work even with large number of ODE terms (during both discovery and prediction)?

Overall, NeuRLP proposed in the paper is novel and significant for the SciML community. Some clarifications are required for its proposed use in the MNN architecture (especially for ODE discovery).

---

### Meta-Review · Area_Chair_TtQE · 2024-03-01

**Recommendation:** Accept (Poster)

**Metareview:**

Authors propose Mechanistic Neural Networks (MNNs) with a differentiable ODE solver NeuRLP for tasks like equation discovery and PDE solving. Some aspects of MNNs need clarification like the model architecture details, comparisons to state-of-the-art methods, and providing more examples of discovered equations. For the camera-ready version, I encourage authors to clarify the MNN components and information flow, provide more implementation specifics on experiments, and further highlight where NeuRLP offers advances over prior differential equation solvers.

---

### Decision · Program_Chairs · 2024-03-02

Accept (Poster)